# LTmatch: A Method to Abstract Pattern from Unstructured Log

**Xiaodong Wang** [1,2], **Yining Zhao** [1,*], **Haili Xiao** [1], **Xiaoning Wang** [1] and **Xuebin Chi** [1,2]

1   Computer Network Information Center, Chinese Academy of Sciences, Beijing 100190, China; xdwang@cnic.cn (X.W.); haili@sccas.cn (H.X.); wxn@sccas.cn (X.W.); chi@sccas.cn (X.C.)
2   Supercomputing Center, University of Chinese Academy of Sciences, Beijing 100049, China
*   Correspondence: zhaoyn@sccas.cn

**Abstract:** Logs record valuable data from different software and systems. Execution logs are widely available and are helpful in monitoring, examination, and system understanding of complex applications. However, log files usually contain too many lines of data for a human to deal with, therefore it is important to develop methods to process logs by computers. Logs are usually unstructured, which is not conducive to automatic analysis. How to categorize logs and turn into structured data automatically is of great practical significance. In this paper, LTmatch algorithm is proposed, which implements a log pattern extracting algorithm based on a weighted word matching rate. Compared with our preview work, this algorithm not only classifies the logs according to the longest common subsequence(LCS) but also gets and updates the log template in real-time. Besides, the pattern warehouse of the algorithm uses a fixed deep tree to store the log patterns, which optimizes the matching efficiency of log pattern extraction. To verify the advantages of the algorithm, we applied the proposed algorithm to the open-source data set with different kinds of labeled log data. A variety of state-of-the-art log pattern extraction algorithms are used for comparison. The result shows our method is improved by 2.67% in average accuracy when compared with the best result in all the other methods.

**Keywords:** log pattern extraction; word matching rate; LCS; log template

## 1. Introduction

Logs are an essential part of computer systems. The main purpose of logging is to record the necessary information generated during the running of programs and systems, and logs are widely used for runtime state recovering, performance analyzing, failure tracing and anomaly detecting. Due to the importance of logs, the vast majority of public released software and systems have certain types of log services. For large-scale applications, such as software and systems running in distributed systems and environments, the generated log files may contain a very large amount of data. The traditional manual method for analyzing such a big number of logs has become a time-consuming and error-prone task. So how to automatically analyze the logs are of great significance for reducing system maintainers' workload and tracing the causes of failures and anomalies.

To achieve the goal of automatic log analysis, many researchers use various data mining methods to analyze logs and diagnose anomalies in recent years. For example, researchers use decision tree, self-encoder, and bidirectional recurrent neural network method based on the attention mechanism to diagnose abnormalities from logs. The above methods have achieved relatively good results in automated log analysis. However, the first step of these automated analysis methods is to classify and structure logs so that features can be extracted and used in different data mining methods. Being a common foundation, the more accurate the classified log categories are, the better the results of the corresponding data mining task would be. Therefore, how to accurately classify logs and extract important information from them becomes very important.

Since developers are accustomed to using free text to record log messages, the original log messages are usually unstructured and flexible. The goal of log pattern extraction is to

classify the unstructured parts of a log and split each of the classified logs into the constant part and variable part. For example, a Linux system log is as follows:

Sep, 30, 03:50:55, HostName, sshd, 309, Invalid user UserName from 0.0.0.0

If separated this log by commas, you can see that the first three fields of the log represent time, and the middle three fields represent the hostname, daemon name, and PID number respectively. These fields as mentioned above belong to the structured part of the log and can be extracted simply using regular expressions, while the log pattern extraction algorithm mainly focuses on the content of the last field, which needs to transform the content of the last field as below:

$$Invalid \quad user < * > from < IP >$$

where the *Invalid user from* is the constant part of the log, and the sign $< * >$ represents the variable part of the log. In the study of log pattern extraction, Zhao [1] propose the algorithm Match, which uses word matching rate to determine the similarity of two logs, and uses this to determine the type of logs. At the same time, they also propose a tree matching algorithm to classify the logs. Later, they propose the Lmatch algorithm [2], which improves the accuracy of the word matching rate algorithm. The algorithm treats each word in the log as a basic unit and then calculates the number of matching words through the longest common subsequence between the logs, and the total words in the two logs are compared to calculate the word matching rate finally. Although the above methods have achieved good results, there are still some aspects that can be improved. First, the word matching algorithm in the paper is too simple and does not take into account the weight information of different parts of a log. Secondly, the original method stores the logs based on the hash table of the first word in the log pattern. Because of the complexity and variability of the log, it is too simple to partition it just by hashing functions. Finally, the parameter adjustment method of the log pattern extraction algorithm is not clear enough.

Based on the above challenges, this article describes the log pattern extraction algorithm LTmatch(LCS Tree match algorithm). This algorithm treats the constant part and the variable part differently when determining the word matching rate of two logs, and uses a weighted matching algorithm. In this way, the detailed changes of the constant part and the variable part of the log in different types of logs can be determined when the logs are matched, so the matching result is optimized. In the storage structure of the log pattern, a tree structure based on log words is used for storage, which further refines the division of the log classification structure. The rules for extracting the log template can distinguish the variable part and the constant part of the log. Moreover, whether the variable part length is changed or fixed can be found by using two different variable symbols. This makes the final template of the log more conducive to subsequent analysis. In the parameter optimization of the entire algorithm, a large number of experiments were carried out to adjust the parameters and the best parameters are determined for a variety of different types of logs. In general, this article has the following two contributions:

First, the method in this paper further optimizes the three points of the workflow, which includes the optimal word matching rate, log warehouse structure, and log template extracting.

Second, the log pattern extraction algorithm performs multi-dimensional experimental analysis on the open-source log datasets, and the experimental results prove the advantages of this method.

In the following, we will introduce related work in Section 2, describe our method in detail in Section 3, conduct experiments in Section 4, and the conclusion is got in Section 5.

## 2. Related Work

In the log anomalies detection area, Chen [3] diagnose the anomalies in large networks through a decision tree. Borghesi [4] use the method of self-encoder to diagnose the anomaly in the high-performance computing system. Meng [5] use two different indicators

in sequence and quantity to detect anomalies in unstructured logs. Zhang [6] use a bidirectional recurrent neural network method based on the attention mechanism to find and diagnose abnormalities from logs. Log parsing is a necessary data preprocessing step to train machine learning models above, so we briefly introduce the research progress of log pattern matching algorithms in different directions below.

### 2.1. A Logging Pattern Extraction Algorithm Based on Frequent Patterns

Vaarandi [7] uses a clustering algorithm called SLCT to classifying log data. This clustering algorithm is based on the Apriori frequent itemset algorithm, so users need to manually input and adjust the support threshold. SLCT will perform two overall scans of the log: the first time the word frequency statistics of all words in the log are performed, and then in the second scan, the pattern cluster of the log is established based on the word frequency obtained in the first scan. After the second scan, the algorithm generates a log template for each cluster based on the established cluster. Subsequently, the author proposes the LogCluster [8] algorithm, which further improves the SLCT so that the results of the algorithm can be more stable when the position of the word constant changes. Nagappan [9] also propose a log pattern extraction method LFA based on frequent pattern mining. The difference between this algorithm and SLCT is that it considers the frequency distribution of words in each log message instead of parsing rare log messages on the entire log data.

### 2.2. Clustering Based Logging Pattern Extraction Algorithm

Qiang [10] uses the LKE method to refine log patterns, which combines clustering algorithms and heuristic rules. There are three steps: The first step is log clustering, a weighted edit distance is used for clustering to measure the distance between two logs, and then use a hierarchical clustering algorithm to cluster the original log messages; the second step is to split the clustering results: perform a heuristic rule-based method to further split the clustering results. The third step is to generate a log template. Tang [11] propose the LogSig algorithm to refine log patterns. The workflow of the algorithm can also be divided into three steps: the first step is to generate word pairs and encode them; the second step is to cluster based on word pairs; the third step is to generate log templates. Mizutani [12] propose the SHISO algorithm to obtain the log template by constructing a structured tree structure. Zhao [1] propose a pattern refinement algorithm Match to determine the similarity of two logs, which is based on the word matching rate of the two logs. Compared with the other algorithms mentioned above, the biggest advantage of this algorithm is that it can be calculated in real-time and does not need to refine the model in advance, which is very conducive to online model refinement analysis. Later, the author improved the Match algorithm in the paper to obtain the Lmatch [2] algorithm. The algorithm improved the one-to-one matching of the fixed positions of the two logs to match according to the longest common subsequence. This can effectively solve the same pattern that two logs have different positions in the constant part.

### 2.3. Log Pattern Extraction Algorithm Based on Other Methods

Salma [13] propose the MoLFI algorithm, which models log analysis as a multi-objective optimization problem and uses evolutionary algorithms to generate log patterns. Jiang [14] propose AEL, a log extraction algorithm based on a hierarchical structure. First, the constant part and variable part of the log are obtained through fixed rules, and then iteratively distinguished according to the number of occurrences of words in the constant part and variable part. However, this method is relatively simple to distinguish between the variable part and the constant part, and there is no consideration of order, which may affect the accuracy of classification. Du [15] propose the Spell, in which a matching rule based on the longest common subsequence is used to get the similarity of logs. By using the structure of the tree, it performed a good advantage for the searching of log templates. Therefore, He [16] propose Drain, which uses a fixed depth tree to save the log templates. As long as

the log length is different, it must be divided into different child nodes, so the fixed depth classification approach cannot distinguish the log that contains the multivariate variables.

Most algorithms based on frequent patterns and clustering-based logging pattern extraction algorithms are offline algorithms because these algorithms need to scan all the historical logs in the first step. However, the proposed method in this article can deal with log parsing online, which is more suitable for production practice. On the other hand, compared with the state-of-art algorithms of other methods such as Drain, our method using a tree structure based on LCS, which can refine the classified result and find the same type of log with different length of words.

## 3. Log Pattern Extraction Algorithm

This section introduces the proposed log pattern extraction algorithm and related optimal. As described in the introduction, the main focus of log pattern extraction is the unstructured part of the log. Therefore, unless otherwise specified, a log described later refers to the content of the unstructured part of the log (for example, the last field: *Invalid user UserName from 0.0.0.0* of the example log in the introduction part). The log pattern extraction algorithm is generally divided into online methods and offline methods. Offline methods usually require historical log data, and then cluster the historical logs through a round of traversal, after that it can extract a template for each type of log. Although this method is intuitive, it cannot form a new category in real-time if a new log does not meet the known category in actual applications. Therefore, the online log classification method is more valuable in the actual application. In general, the flow of the online log classification algorithm is shown in Figure 1.

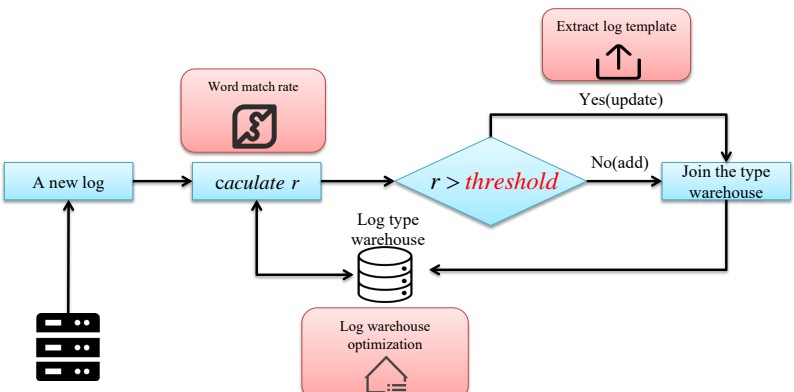

**Figure 1.** The Flow Chart of Online log classify.

As shown in the picture above of the overall process, a log is obtained from the server, and then the word matching rate is calculated with the existing log in the log type warehouse. If the result is greater than the threshold, it is considered to belong to the category, and then the template of the category is updated. If on the opposite, add this log to the log warehouse as a new type. It can be seen from this process that there are three key issues that need to be solved in the entire process. One is how to determine the word matching rate of the two logs. The second is how to construct a log warehouse that is conducive to store or search the log patterns. The third is how to extract the log template. Therefore, we will introduce the word matching rate in Section 3.1, the log warehouse in Section 3.2, the log template extraction method in Section 3.3. In the Section 3.4, we will perform an example to show the overall process.

### 3.1. The Word Matching Rate of Two Logs

3.1.1. Basic Word Matching Rate Algorithm

Regarding the question of how to determine whether two logs belong to the same pattern, the most intuitive method is to determine whether the two logs match enough

words, so the word matching rate can be used to determine the log pattern. The algorithm treats each word in the log as a basic unit and then matches it. Specifically, assuming that the original log is $l$, the log to be matched is $l'$, and the number of words contained in the two logs is $m$ and $n$ respectively, the word matching rate calculation formula for the two logs is as follows:

$$r(l,l') = \frac{|Match(l,l')| \times 2}{m+n} \tag{1}$$

where $|Match(l,l')|$ represents the number of one-to-one matching words in the corresponding positions of the two logs. However, if the word matching rate algorithm based on Formula (1) is used, it will bring a side effect. That is, when the number of words in the variable part of the log is uncertain, the constant part of the two logs will be misaligned as the constant part is matched, which means the Match algorithm will fail. In order to make up for this defect, the improved calculation formula for word matching rate is as follows:

$$r(l,l') = \frac{|LCS(l,l')| \times 2}{m+n} \tag{2}$$

where $|LCS(l,l')|$ represents the number of words matched when the two logs are calculated using the longest common subsequence. The algorithm that uses this word matching rate for pattern refining is called the LMatch algorithm.

### 3.1.2. Optimized Word Matching Rate Algorithm

Further analysis of Formula (2) can find that if the 2 of the numerator in Formula (2) is moved to the denominator, the denominator becomes the harmonic average of the length of the original log and the log to be compared. According to actual processing experience, it can be found that the number of constant parts and variable parts of different types of logs will be slightly different when the logs are matched. Therefore, to improve the calculation accuracy, the optimized word matching rate algorithm further multiplies the length of the two logs by an adjustment weight. According to this idea, Formula (2) is improved, and the calculation formula of the final word matching rate algorithm is as follows:

$$r(l,l') = \frac{|LCS(l,l')|}{wm + (1-w)n} \tag{3}$$

where $w$ represents the adjustment weight, which is a parameter that needs to be adjusted according to different training sets in practice. The algorithm that uses Formula (3) to refine the pattern is called the LTMatch algorithm.

### *3.2. Log Type Warehouse*
### 3.2.1. Basic Log Type Warehouse

The basic unit of log type warehouse is a log template string containing constant words and variable wildcards. In the actual process to construct the warehouse, whenever there is a new log for matching, the necessary step is to use this new log data and all log template strings to calculate the word matching rate. Therefore, the basic log type warehouse can directly use the list for storage, so that when calculating the word matching rate, only a simple cycle of the list is required to complete the matching calculation. The number of iterations during the cycle is equal to the number of log patterns in the warehouse.

### 3.2.2. Optimized Log Type Warehouse Storage Structure

When storing the logs based on the basic log type warehouse, it can be found that it is necessary to calculate the longest common subsequence of the log and all known log patterns separately whenever a new log appears. As the log pattern increases, the scale of the computation will become larger and larger, so if the number of comparisons can be reduced, the overall efficiency of the algorithm can be significantly improved. The optimized log type warehouse aims to solve this problem.

On the optimization method of log warehouse, this article is inspired by Drain [16]. The log type warehouse in the "LTmatch" algorithm uses a storage form based on a tree structure, in which the node of each tree stores the word at the corresponding position of the log. If a number appears in the word, it will be stored in the tree node starting with the wildcard <*>. Different from the log store structure of Drain, the first layer of Drain (the root node is considered to be the 0th layer in this article) stores the number of logs. The top-level classification of the tree cannot be performed by fixing the length of the log because the optimized word matching rate algorithm is based on the longest common subsequence(LCS). The LCS method can find different lengths of the logs that belong to the same category naturally. Therefore, the tree structure we used starts with a node of depth 1 (the depth of the root node is 0 in this article). By analogy, the node with depth n stores the nth word of the log. A schematic diagram of a log pattern warehouse constructed in this way is shown in Figure 2.

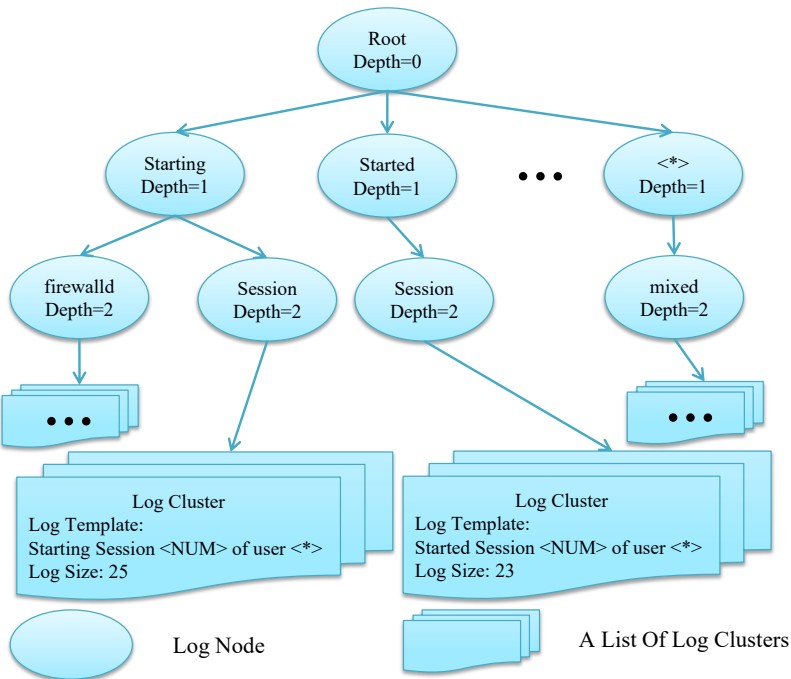

**Figure 2.** Figure of Log Patterns Warehouse.

The depth of the log warehouse tree structure constructed in Figure 2 is two. Because the leaf node stores a list of log clusters in practice, which is different from the structure of other nodes in the tree, so it does not add the value of the parameter "depth" in the program. Specifically, all of the branch nodes that construct the tree structure contain a depth parameter and a word. In particular, the depth of the root node is 0, the word "Root" is used to represent the root node, and the other branch nodes store the words that represent the actual log. Same as the method Drain in the paper, if a number appears in a word or the number of nodes in this layer exceeds a predefined value of the parameter "max_children", the log template will be uniformly located at a special node "<*>" in the current layer. The leaf node of the log warehouse tree structure stores a list, each element is represents a log cluster structure. The structure contains a log template representing the log cluster and a size variable to save the corresponding log number which is the occurrences of the log type.

In the actual construction of the log warehouse tree structure, the matching rules, and template extraction process based on LCS may change the words of the template, which will lead to the unstable position of the log template in the tree structure. To solve this problem, a template extraction algorithm based on tree depth is used in template extraction. Before getting the log template, the function read the depth parameter first.

Then keep the previous depth word unchanged when extracting the template. The specific algorithm is shown in the following Section 3.3.2. This calculation mode is also more reasonable for the log because the variables of the log usually appear in the back position.

According to the storage mode of the tree structure in Figure 2, it can be seen that as long as the log warehouse is saved according to the storage format. When a new log comes, it can be matched to the corresponding subset of the log pattern warehouse according to the information at the beginning of the log. Then calculate the LCS between this log and all the log templates in the list of the subset, which greatly reduces the number of log comparisons.

### 3.3. Log Template Extraction Method

3.3.1. Basic Log Template Extraction Algorithm

The goal of the log template extraction algorithm is to get the constant part of a log and use a special symbol to represent the variable part. Therefore, after obtaining the LCS of the logs, the basic log template extraction algorithm only needs to traverse each word in the LCS once. See Algorithm 1 for detail.]

---

**Algorithm 1:** Log Template Extraction Algorithm

---

**Input:** log template in warehouse $l_m$, compared log $l$
**Output:** log template to return *retVal*
1: **Initialize: set** $retVal = [\,]$
2: $lcs \leftarrow LCS(l_m, l)$
3: $lcs.reverse()$
4: **for** $i = 1, 2, ..., length(lcs)$ **do**
5: 　**if** $l[i] == lcs[lenght(lcs) - 1]$ **then**
6: 　　$retVal.append(l[i])$
7: 　　$lcs.pop()$
8: 　**else**
9: 　　$retVal.append(<*>)$
10: 　**end if**
11: 　**if** $length(lcs) == 0$ **then**
12: 　　break
13: 　**end if**
14: **end for**
15: **if** $i < length(l)$ **then**
16: 　$reVal.append(<*>)$
17: **end if**

---

It can be seen from Algorithm 1 that the log template extraction algorithm does not consider whether the first few characters of the log are variables or not because the algorithm corresponds to the basic log template warehouse, which stores all log templates in a list structure, so regardless of the change of the log template will not affect the position of the template in the list. The algorithm only pays attention to the correspondence between the LCS of the two logs and the different positions of the log to be matched when performing matching.

3.3.2. Optimized Log Template Extraction Algorithm

In the basic log template extraction Algorithm 1, the log template for the basic log type warehouse can be obtained. However, as described in Section 3.2.2, when the tree structure of the log storage warehouse is used, in order to ensure the stability of the log template, the basic version of the log template extraction algorithm needs to be improved, and the improved log template extraction algorithm needs to ensure the preview depth length of the word at the position does not change. On the other hand, considering that the advantage of LCS is that the constant part of the log can be accurately obtained, the change of the variable part of the word can be obtained through the precise comparison of the

constant part. Therefore, in order to make full use of the advantages of LCS, the improved log template extraction algorithm should be able to further confirm whether the variable corresponds to a single variable or a multivariate variable.

In order to meet the above requirements, when the log template extraction algorithm is designed, the *depth* parameter needs to be considered first. During the construction of the log template, if the loop has reached depth, end the loop early and add the first *depth* words of the logs to be compared directly to the log template. On the other hand, in order to make the information of the final template not only distinguish between variables and constants but also to further refine the variables, the variable part of the algorithm uses two kinds of special signs. One is "<*>" to express the single variable, the other one is "<+>" to express the multivariate variable. In practice, it is necessary to calculate the longest common subsequence of the two logs to be matched first, and then calculate the lengths of non-matching positions which decision the kinds of special signs to be replaced. If the number of variables between two constant words is greater than or equal to 2, it means that this position is a multivariate variable, so use the special sign <+> for description. Specifically, the log pattern refining algorithm flow is summarized as shown in Algorithm 2.

---

**Algorithm 2:** Log Template Extraction Algorithm Based On Tree Structure

---

**Input:** log template in warehouse $l_m$, compared log $l$, the depth of tree *depth*
**Output:** log template to return *retVal*
1: **Initialize: set** $retVal = []$
2: $lcs \leftarrow LCS(l_m, l)$
3: **if** $length(lcs) < depth$ **then**
4:     **return** *retVal*
5: **end if**
6: **for** $i = length(l) - 1, length(l) - 2, \ldots, depth$ **do**
7:     **if** $length(lcs) == 0$ **then**
8:         **break**
9:     **end if**
10:     **if** $l[i] == lcs[length(lcs) - 1]$ **then**
11:         $retVal.append(l[i])$
12:         $lcs.pop()$
13:     **else**
14:         **if** $i == length(l) - 1$ **then**
15:             $retVal.append(<*>)$
16:         **else if** $retVal[length(retVal) - 1] ==<*>$ **then**
17:             $retVal.pop()$
18:             $retVal.append(<+>)$
19:         **else if** $retVal[length(retVal) - 1] ==<+>$ **then**
20:             **continue**
21:         **else**
22:             $retVal.append(<*>)$
23:         **end if**
24:     **end if**
25: **end for**
26: $retVal.reverse()$
27: $retVal = l[: depth] + retVal$

---

According to the Algorithm 2, the template of any two logs can be obtained. The advantage of this algorithm is that it not only increases the information of the log template but also makes the word matching rate algorithm of the log more reasonable because the length of the log template is used in the word matching rate algorithm. Obviously, the position of the multivariate variable in the length should not reflect different length information under the change of the variable length.

Finally, the entire online log type classification algorithm process represents the Lmatch algorithm in the subsequent experiments if the basic algorithm of log word matching rate, log type warehouse, and log template extraction algorithm in the previous sections are brought into Figure 1. The improved algorithm LTmatch proposed in this paper is obtained by compositing the three optimizations into the process shown in Figure 1.

### 3.4. Example Procesure of the LTmatch Algorithm

In this subsection, we will display an example to explain the whole process of the log parsing proceeding. Before feeding the logs to the log parsing algorithm, all the logs will be preprocessed by replaced the frequent signs with a special token surrounded with "<>". Assuming there are three logs will be fed into the log parsing algorithm. The original logs and preprocessed logs are shown in the second column and the third column in Table 1.

**Table 1.** The Example logs to be processed.

| Log Name | Original Logs | Preprocessed Logs |
|----------|---------------|-------------------|
| Log1 | Failed password for invalid user UserNameA from 121.132.1.20 port 51,469 ssh2 | Failed password for invalid user UserNameA from <IP> port <NUM> ssh2 |
| Log2 | Failed password for root from 40.127.204.124 port 42488 ssh2 | Failed password for UserNameB from <IP> port <NUM> ssh2 |
| Log3 | Starting Session 122985 of user UserNameB | Starting Session <NUM> of user UserNameA |

First, Log1 comes to the online log classification algorithm, the algorithm will search the **Log warehouse,** then it finds the Log warehouse is empty. So this piece of the log will be inserted into the log warehouse according to the *depth* parameter in Algorithm 2. Assuming we set the *depth* to 2. The log will be inserted to the node of "password". after that, the log warehouse as shown in Figure 3a.

Second, the log2 comes, the algorithm will search the log warehouse by the first word and the second word of log2, and then it finds that the "Failed" node and "password" existed in the log warehouse. So the log2 will calculate the **word matching rate** with all the Log Clusters below the node "password" in turn until the result is bigger than the preset threshold. In this example, there is only one Log Cluster below the node "password", so Log2 will calculate the word matching rate with the LogCluster in Figure 3a. Assuming the threshold is 0.45 and the weight is 0.4, the first three words and the word "from" of the Log1 and Log2 match, and Log1 has eleven words, Log2 has six words. Therefore, according to Formula (3), we can calculate r(log1, log2) = 4/(0.4*11 + 0.6*6) = 0.5 > 0.45. So the two logs belong to one class. Then, the log cluster will be updated. To clearly the **Log Template Extraction Method**, we display the two logs to be updated below:

Failed password for invalid user UserNameA from <IP> port <NUM> ssh2
Failed password for UserNameB from <IP> port <NUM> ssh2

If we use Algorithm 1, the template of two logs above will be the *Failed password for <\*> <\*> <\*> from <IP> port <NUM> ssh2*. This result is because the variable is too long to make three <\*> add to the template. If we use the algorithm 2, the result of template will be the *Failed password for <+> from <IP> port <NUM> ssh2*. This result will make the template more simple and reasonable. After that, the log warehouse as shown in Figure 3b.

Third, Log3 comes to the online log classification algorithm, the algorithm will search the Log warehouse, then it finds there is not node "Starting" in the log warehouse. So this piece of the log will be inserted into the log warehouse by constructing the node "Starting" and node "Session". Then the log warehouse as shown in Figure 3c.

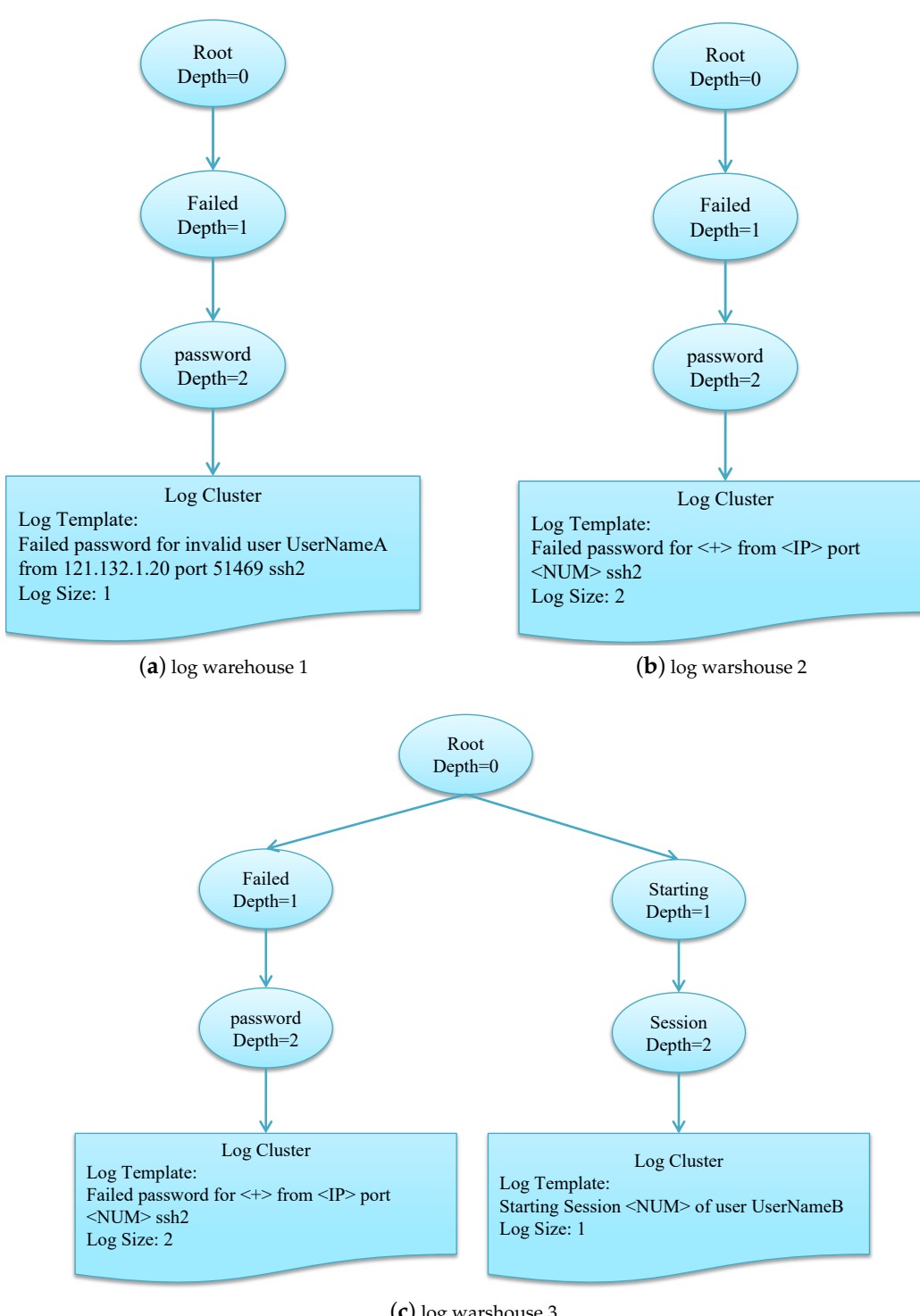

**Figure 3.** The Example for constructing A Log Warshouse.

Following the process described above, logs are added to the log warehouse one by one. We will prove the advantages of this algorithm through experiments in the next section.

## 4. Experiments

In this section, the accuracy and robustness advantages of the LTmatch log parsing algorithm are determined by experimenting with 16 different log types of data sets and comparing them with a variety of the most advanced log pattern extraction algorithms.

### 4.1. Dataset

The real-world log data is relatively small because the log information of companies and scientific research institutions usually contains the private information of users. Fortunately, Zhu [17] published an open-source data set Loghub [18] in the paper, which contains 16 Logs generated by different systems and platforms. It contains a total of more than 40 million logs, with a capacity of 77 G. For specific log content introduction, please refer to the original paper. In our experiment, we divided these public data into two data sets and then conducted log classification research experiments. The detailed introductions are as follows:

Dataset 1: The sample published by the author of the paper on the GitHub platform is selected. In this sample, 2K logs are randomly selected for 16 types of logs, and then manually marked by professionals, and the corresponding log pattern is obtained.

Dataset 2: The Loghub dataset log was published by the author of the paper, in which 450,000 logs are selected from the BGL and HDFS logs respectively, and then the real results are labeled according to the log template.

### 4.2. Evaluation Methods

In order to verify the method in this paper as detailed as possible, we conduct experiments through the following evaluation methods and rules.

Accuracy: The ratio of correctly resolved log types to the total number of log types. After parsing, each log message corresponds to an event template, so each event template will correspond to a set of log messages. If and only if this group of messages is exactly the same as a group of log messages corresponding to the real artificially tagged data, the parsing result of the log pattern is considered correct.

To avoid random errors caused by the experiment, we calculated the results of each group of experiments ten times and obtained the average value. For fairness of comparison, we apply the same preprocessing rules (e.g., IP or number replacement) to each log extracting algorithm. The parameters of all the log extracting algorithms are fine-tuned through grid search and the best results are reported. All experiments are performed on a computer with 8 G memory, model Intel(R) Xeon(R) Gold 6148, and CPU clocked at 2.40 GHz.

### 4.3. The Accuracy of Log Extracting Algorithms

This section conducts the experiment on the accuracy of the log pattern extraction algorithm. The experiment selected dataset 1 as the benchmark, and the comparison methods were the five methods with the highest average accuracy shown in Zhu [17]. The experimental results are shown in Table 2. The first five columns in the table are five methods for comparison, and the methods in the last two columns are the methods proposed in this paper. In the contrast method, LenMa uses the clustering algorithm in traditional machine learning to determine the log type. The Sepll algorithm uses the longest common subsequence based on the prefix tree to determine the log type. The difference from the prefix tree in this paper is that the prefix tree is used to store the characters of the longest common subsequence that has appeared. The AEL algorithm uses a heuristic algorithm to get the log type. IPLoM uses iterative splitting technology to fine-grain the types of logs. Drain uses a fixed depth tree to store the log structure. The first layer of its storage tree stores the length of the log, so when comparing, it is guaranteed that each log to be compared is a log of the same length.

In the results shown in Table 2, each row represents the type of experiment log, and each column represents a different log parsing method. To make the results clearer, the last row in the table (except the last result) shows the average accuracy of the method. The last column in the table is the best accuracy calculated in each log dataset, and the last value is the best average accuracy. For each dataset, the best accuracy is token by a sign "*" and the best accuracy values are also shown in the last column of the table. In particular, the accuracy of all results greater than 0.9 is marked in bold, because these results can be considered to be excellent on these datasets.

**Table 2.** The average accuracy of the results on different kinds of log.

| Log Name | LenMa | Spell | AEL | IPLoM | Drain | LTmatch | Lmatch | Best |
|----------|-------|-------|-----|-------|-------|---------|--------|------|
| Andriod | 0.8795 | **0.9195 *** | 0.6815 | 0.7120 | **0.9110** | 0.9080 | **0.9120** | 0.9195 |
| Apache | **1.0000 *** | **1.0000 *** | **1.0000 *** | **1.0000 *** | **1.0000 *** | **1.0000 *** | **1.0000 *** | 1.0000 |
| BGL | 0.8295 | 0.7545 | **0.9570** | 0.9390 | **0.9625** | 0.9325 | 0.8355 | 0.9625 |
| Hadoop | 0.8850 | 0.7840 | **0.9690** | 0.9540 | 0.9475 | **0.9870 *** | 0.8765 | 0.9870 |
| HDFS | **0.9975** | **1.0000** | **0.9975** | **1.0000** | 0.9975 | **1.0000** | 0.8405 | 1.0000 |
| HealthApp | 0.1740 | 0.6390 | 0.5675 | 0.8215 | 0.7800 | 0.8795 | 0.5565 | 0.8795 |
| HPC | 0.8295 | 0.6540 | **0.9030** | 0.8290 | 0.8870 | **0.9345** | **0.9040** | 0.9345 |
| Linux | 0.7010 * | 0.1505 | 0.6725 | 0.6715 | 0.6900 | 0.6905 | 0.5995 | 0.7010 |
| Mac | 0.6980 | 0.1505 | 0.6725 | 0.6715 | 0.6900 | 0.6905 | 0.5995 | 0.7010 |
| OpenSSH | **0.9250 *** | 0.5540 | 0.5380 | 0.5400 | 0.7875 | 0.7475 | 0.6780 | 0.9250 |
| OpenStack | 0.7425 | 0.7535 | 0.7575 | 0.3305 | 0.7325 | 0.8595 * | 0.8065 | 0.8595 |
| Proxifier | 0.5080 | 0.5265 * | 0.4950 | 0.5165 | 0.5265 * | 0.5265 * | 0.5165 | 0.5265 |
| Spark | 0.8835 | **0.9050** | **0.9050** | 0.9200 | 0.9200 | **0.9975 *** | 0.7175 | 0.9975 |
| Thunderbird | **0.9430** | 0.8435 | **0.9410** | 0.6630 | **0.9550 *** | 0.9470 | **0.9215** | 0.9550 |
| Windows | 0.5655 | 0.7055 | 0.6895 | 0.5670 | **0.9970 *** | 0.9960 | 0.9900 | 0.9970 |
| Zookeeper | 0.8405 | 0.8180 | **0.9210** | 0.9615 | 0.9665 | **0.9885 *** | 0.9640 | 0.9885 |
| Average | 0.7751 | 0.7336 | 0.7912 | 0.7562 | 0.8654 | 0.8885 * | 0.8103 | 0.8885 |

* The best accuracy is highlighted with a asterisk and shown in the column "Best" for each dataset.

From the results of Table 2, we can find out that LTmatch has achieved the best average accuracy, and the algorithm also contains the most optimal accuracy. Compared with the Lmatch algorithm before optimization, the average accuracy is improved by 9.65%, which shows that the fine-grained division of the tree structure can improve the accuracy of log classification. Compared with other log parsing algorithms, the LTmatch algorithm we proposed is 2.67% better than the best algorithm Drain among the other methods. It proves that the algorithm in this paper can more accurately identify the log type by comparing the longest common subsequence based on the weight.

### 4.4. Robustness

In this section, in order to verify the robustness of the method we proposed, we use the method of LTmatch to conduct accuracy experiments on log datasets of different volumes, so this experiment chooses the labeled logs in dataset 2 as the experiment object. In comparison, the same method as the previous section was selected for the experiment, and the final experimental result is shown in Figure 4.

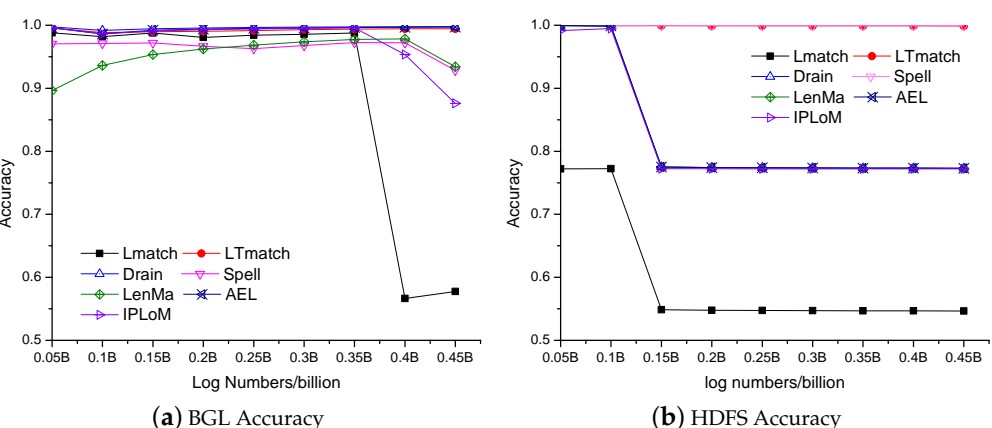

**Figure 4.** Accuracy of different log parsers over different sizes of BGL/HDFS log data sets.

First, the comparison between the optimization method we proposed and the original method is analyzed. It can be seen from Figure 4 that the initial results of Lmatch are better on HDFS and GBL data, but after a certain amount of logging, the accuracy of the

results begins to decrease significantly, which shows that directly using the word matching rate algorithm based on the longest common subsequence has no obvious advantage in parsing log patterns. The accuracy of LTmatch has remained stable when the log capacity changes, indicating that the improved log pattern extraction algorithm has more strong generalization ability than the algorithm before the improvement.

When comparing with all other types of log classification algorithms, it can be seen from the Figure 4a on the GBL dataset, the initial accuracy of LenMa algorithm, IPLoM algorithm, and Spell algorithm is relatively high, but they have begun to decline when the number of logs reaches 0.45 million. The remaining Drain, AEL, and the LTmatch algorithm we proposed have maintained stable accuracy. On the HDFS dataset in Figure 4b, except for Spell and the LTmatch algorithm we proposed, the accuracy has been maintained at a high level, and other methods have begun to decline on the number of logs of 0.15 million. It can be seen from the experimental results of the two different log datasets HDFS and GBL on different data volumes that only the LTmatch algorithm can always remain stable when the log type changes. The rest of all other types of log classification algorithms cannot keep a stable accuracy of log classification in different volumes of the logs from multiple datasets. Therefore, the generalization ability of the log classification method we proposed is the best among all the state-of-art methods.

### 4.5. Efficiency Analysis

In order to analyze the efficiency of the algorithm, this article uses the same dataset as the previous section and uses the same method to classify patterns and record the time on the logs that are split into the same scale. The final experimental results are shown in Figure 5. It should be noted that in order to make the result clearer, the maximum time period is set to 145 s in Figure 5a, so the LenMa algorithm is not fully displayed, but the trend of increasing time is already obvious.

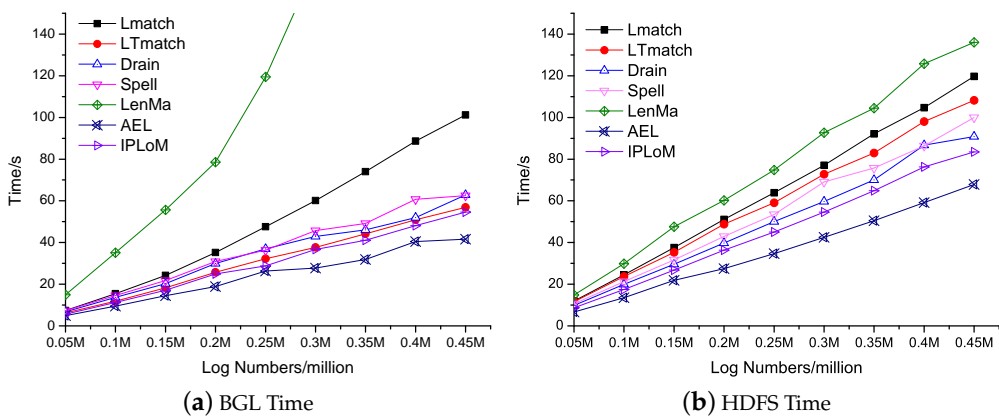

(**a**) BGL Time          (**b**) HDFS Time

**Figure 5.** Time of different log parsers over different sizes of GBL/HDFS log data sets.

First, the improved algorithm LTmatch in this paper is compared with the original algorithm Lmatch. It can be seen that the LTmatch algorithm has improved efficiency in both GBL data and HDFS data, which shows that the optimized algorithm introduces a tree structure for log type storage increases the complexity of the container of the log templates, while it has a better advantage of efficiency.

Compared with all other algorithms, on the GBL dataset, except for the LenMa algorithm, which has a faster increase in time consumption, other algorithms have linear growth. The efficiency of the LTmatch algorithm we proposed can reach the top three. On the HDFS data set, although the algorithm we proposed is relatively backward, it has always been relatively stable and increases linearly. This is because the time complexity of the LTmatch algorithm is $O((d + cm_1m_2)n)$, where $d$ is representing the depth of the parse tree, and $c$ is the number of log templates contained in the leaf nodes currently searched. $m_1$ and $m_2$ are the number of words in the two logs that were matched separately. Obviously,

$d, m_1, m_2$ are constants. $c$ is also a constant compared to the increase in the number of logs when the overall algorithm is in progress. Moreover, the number of templates itself is much smaller than the total number of logs, and at the same time, after the tree is divided, the number of templates for each node is even less. Through the above discussion, it can be seen that the LTmatch algorithm has a level of $O(n)$ time complexity. The actual situation of time increase in Figure 5 is also consistent with it. Therefore, the analysis efficiency of LTmatch is acceptable under the condition of ensuring the highest accuracy.

*4.6. Discussion*

In this section, we will further discuss the rationality of the LTmatch algorithm proposed in this article.

In order to illustrate the rationality of the weight-based log word matching rate design in the LTmatch algorithm, the ratio of the number of constants and variables in log templates of all the templates in the log dataset 1 is counted. The results are shown in Table 3. According to the Table 3, it can be seen that the variable part of the log template is generally less than the length of the constant part, which is very common sense, because in most cases, the variable of the log only records the key change information of the program during operation, and the longer length of the constant part is helpful for the readability of the log. However, the constant part and the variable part of different logs have certain differences, indicating that the proportion of different types of logs corresponding to templates will have certain differences in practice. Therefore, when designing the word matching rate algorithm, the weight is designed so that the algorithm can learn the required weight from the characteristics of the log, thus improving the overall accuracy of the algorithm.

**Table 3.** The Ratio of The Number of Constants and Variables in Log Templates.

| Log Name | Total Words Number | Constant Tokens | Variable Tokens |
| --- | --- | --- | --- |
| Andriod | 887 | 93.69% | 6.31% |
| Apache | 38 | 78.95% | 21.05% |
| BGL | 1141 | 92.02% | 7.98% |
| Hadoop | 814 | 96.44% | 3.56% |
| HDFS | 102 | 96.08% | 43.92% |
| HealthApp | 322 | 89.44% | 10.56% |
| HPC | 253 | 90.12% | 9.88% |
| Linux | 666 | 92.49% | 7.51% |
| Mac | 3503 | 93.58% | 6.42% |
| OpenSSH | 242 | 88.84% | 11.16% |
| OpenStack | 354 | 90.40% | 9.60% |
| Proxifier | 100 | 92.00% | 8.00% |
| Spark | 229 | 91.27% | 8.73% |
| Thunderbird | 846 | 88.89% | 11.11% |
| Windows | 377 | 92.57% | 7.43% |
| Zookeeper | 368 | 85.87% | 14.13% |

## 5. Conclusions

Log pattern extraction algorithm is critical for downstream log analysis tasks. This paper proposes an online log pattern extraction algorithm of LTmatch. This algorithm is improved by applying a weight-based log word matching rate and the hypothetical space of log classification through reasonable settings. In the log template, the LTmatch distinguishes between multivariate variables and unit variables in log variables and uses a tree structure to optimize the storage of log patterns.

We set up multiple sets of experiments to verify the accuracy and robustness of the method in this paper. The result shows that the LTmatch algorithm gets the best accuracy compared with other state-of-the-art log pattern extraction methods, which means LTmatch

is a good method. Also, LTmatch shows the best average accuracy in 16 different kinds of logs, which means LTmatch has a good generalization ability.

**Author Contributions:** Funding acquisition, H.X. and X.C.; Investigation, X.C.; Project administration, H.X.; Supervision, X.W. (Xiaoning Wang); Writing—original draft, X.W. (Xiaodong Wang); Writing—review & editing, Y.Z. All authors have read and agreed to the published version of the manuscript.

**Funding:** This research was funded by the Strategic Priority Research Program of the Chinese Academy of Sciences, Grant No. XDA19020101.

**Acknowledgments:** We would like to thank the financial support from the Strategic Priority Research Program of the Chinese Academy of Sciences, Grant No. XDA19020101.

**Conflicts of Interest:** We declare that we have no financial and personal relationships with other people or organizations that can inappropriately influence our work, there is no professional or other personal interest of any nature or kind in any product, service and company that could be construed as influencing the position presented in the manuscript entitled.

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
