# Peer review of "LTmatch: A Method to Abstract Pattern from Unstructured Log"

_applsci, doi:10.3390/app11115302_

Round 1

Reviewer 1 Report

I would like to point out that this is interesting work and worth purusuing, as the authors indicate, the use of logs for analisys is important.

Regarding the Introduction:

I do not really get why you start citing a paper that is 17 years old [1] unless it its a breakthrough article, that sets the tone from that point on. You continue citing different work [2-5] in an attempt give some related work in the Introduction. Unless you are situating your work within a specific context, I do not really see the point. I think that it is better for the reader to know where your work stands regarding the different techniques, not specific works in the introduction. If you want to have this comparison directly with other work, I would appreciate as reader to find this comparison in another section be it related work or background (which you already have).

The related work section, reads well and is structured in a way the reader can understand a bit where the research in the area of log processing is going. I think in this section authors should present where their work stands in relation to the different tecniques and highlight their contribution that is presented in section 3.

The structure in 3 is not clear, you say that you identify 3 key issues

1) word matching

2) extraction

3) warehouse construction

which leads to thinkt that

3.1) word matching

3.2) warehouse and

3.3) extraction method

will deal with this points but the order of the section is not coherent or in the same order as the key points explained. Is this order not relevant or do they not have a dependence of a previous one?

Before dealving into the actual contribution I would like to see the text with a structure that is easy to follow.

Regarding the results, there is a interesting statement saying that the proposed LTmatch method is the best in terms of generalization among all state of the art methods, yet in the conclusions you say that you get relatively good performance. 

Could this sections be complete in the sense that you can state not only about the performance but as you say the ability of generalication and how you stand in relation with other work. It would be good to read a more complete  conclusion section that gives you a good summary of what you find in the presented work.

Reviewer 2 Report

In this paper the authors propose a new “LTMatch” algorithm to extract log patterns from common open-source software logs. IN doing so, they maintain the “log pattern warehouse” – a storage of log patterns which is used in the extraction and classification process. Authors build upon their previous work and improve their “LMatch” algorithm. This version is, of course, considerably better than their last version and 2.67% better (on average) than the state-of-the-art.

Major objections are as follows:

  • Bad presentation – the algorithms are hard to follow. I suggest you use a running example and provide a better, more user-friendly explanation of the procedure.
  • Please comment on the word ordering when matching logs. Do you take word order into account? What is the difference? E.g. if we compare “Today is a nice day”, “Nice day today” and “Today is not a nice day”.
  • Please comment on optimizing algorithm (hyper)parameters. E.g. you say “Where w represents the adjustment weight, which is a parameter that needs to be adjusted according to different training sets in practice.” but do not comment on that when you compare your algorithm with other algorithms using various different datasets (there is only this vague sentence at 421: “Therefore, when

designing the word matching rate algorithm, the weight is designed so that the algorithm can learn the required weight from the characteristics of the log, thus improving the overall accuracy of the algorithm.”).
Can the 3rd party algorithms also be optimized? Did you make a fair comparison then? 

  • Bad English. The paper really needs to be proof-read, as it varies in quality and sometimes has a downright bad sentence, eg:
    • “The inner mark of the form with * is the result means which is the same method as the best accuracy value.“
    • “During the cycle of the log template, if the depth has been cycled to the position, the cycle will be terminated early.”
    • “Although the above methods have achieved relatively good results, some aspects of which can be

improved.”

  • “For the log templates are complex and changeable, it is too simple to divide them only by the

hash function.”

  • “The improved algorithm LTmatch proposed in this paper can be got by adding the optimized algorithm of the above three parts to the process of Fiure 1.” (also, note the typo on Figure 1)
  • “…, so the fixed depth makes the result of classification cannot distinguish the log which contains the multivariate variables.”
  •  

Also, it seems that you are repeatedly using “character” when you should be using “word” (175, 176, 258, 260) which is particularly worrying since that is the focus of your work not just in this paper!?

Minor objections:

  • On line 47 I’d make it clear it is your work.
  • Fig1: what is the meaning of the rocket? Is t(threshold) a function? Please comment further.
  • Table 1: give accurate title, eg. “The average accuracy…”, explain bold and asterisk before table is presented. Why are there two chapters: accuracy and efficiency when there is only accuracy analysis (table 1) which is commented in “efficiency analysis” chapter? Also, in Table1, provide the number of items parsed so that we get a sense of the log files’ size.
  • Chapter 4.6. is again “4.6. Efficiency Analysis”, the same as “4.4. Efficiency Analysis”
  • In 4.5. why did you use only BGL and HDFS? Please expand the analysis to all logs.
  • Fig 3 is not readable, please adjust the graphic (you might try zooming in on the y axis to eg 0.5-1.0)
  • Format table 2. See how to align numbers and text. I suggest adding a separate column with the absolute number of tokens, and then present only percentages in constant and variable tokens columns
  • Expand conclusion.

Round 2

Reviewer 1 Report

I see that the authors have improved greatly the writing of the paper and know its more clear the exposition of the work.

While the main work seems to have enough merit and soundness,  I still find the work has writing issues.

I would suggest to  fix them before publishing this work. Examples notwithstanding, please perform a complete check of the writing of  this work:

Line 9: deep tree to store, ( this begs to answer the question, to store what? )

Line 169: ... to subsequent dealing. ( This is not at all clear what you mean here. )

Line 448: improvemented ( I dont think this work is correct or that exists)

Lines 454 and 455. (When you refer to LTmatch you talk as if it where first or second person with have and show. I am not an English native speaker but I think it should be LTmatch has and shows in the third person. )

Reviewer 2 Report

I am sorry, I still have two major concerns:

  • The running example is “not running”, you have only added the minimal example early on in your processing pipeline. You should have an example that runs through all the stages. Be kind to the reader – make the paper easier to follow.
  • I am not a fan of grammar/style elitism and I am not a native speaker myself, but you simply must invest in the extensive editing of English language and style. I suggest you find a dedicated service and reformat your paper to meet the threshold and standards of the journal.
    g. in you conclusion (which is too short, BTW) you say “This algorithm is improvemented”!?

Then “it distringuishes” followed by “use” and not “uses”. Then “LTMatch show” instead of “shows”, etc.

Also, when you say in the conclusion:

“The result shows that LTmatch algorithm get the best accuracy compared with other state-of-the-art log pattern extraction methods, which means LTmatch have a good performance.”

You are equating accuracy and performance, which is unusual and I think you should be more careful in your terminology.

The conclusion should really be heavily refactored.

Round 3

Reviewer 2 Report

The algorithms 1 and 2 should also include the running example.

I did not see "the details of the modifications" since they are not highlighted and I'm surely not going to diff the entire paper with the previous one (or am I missing some additional file?).

As for the english language, I feel it is still unacceptable, let's take the abstract as an example:

====

Logs record valuable data from different software and systems, and the necessity of analyzing logs has been increasingly significant. Due to the unstructured contents of logs, it needs to categorize logs into log patterns in order to retrieve information and carry out further analyses.

===

etc.

I'm going to leave the verdict to the editor.
